# Acute and endothelial-specific Robo4 deletion affect hematopoietic stem cell trafficking independent of VCAM1

**Stephanie Smith-Berdan**[1,2], **Alyssa Bercasio**[1©], **Leah Kramer**[1©], **Bryan Petkus**[1©], **Lindsay Hinck**[1,3], **E. Camilla Forsberg**[1,2]*

**1** Institute for the Biology of Stem Cells, University of California-Santa Cruz, Santa Cruz, CA, United States of America, **2** Department of Biomolecular Engineering, University of California-Santa Cruz, Santa Cruz, CA, United States of America, **3** Department of Molecular, Cell and Developmental Biology, University of California-Santa Cruz, Santa Cruz, CA, United States of America

© These authors contributed equally to this work.

* cforsber@ucsc.edu

**Data Availability Statement:** All relevant data are within the manuscript and its Supporting Information files.

## Abstract

Hematopoietic stem cell (HSC) trafficking is regulated by a number of complex mechanisms. Among them are the transmembrane protein Robo4 and the vascular cell adhesion molecule, VCAM1. Endothelial VCAM1 is a well-known regulator of hematopoietic cell trafficking, and our previous studies revealed that germline deletion of Robo4 led to impaired HSC trafficking, with an increase in vascular endothelial cell (VEC) numbers and downregulation of VCAM1 protein on sinusoidal VECs. Here, we utilized two Robo4 conditional deletion models in parallel with Robo4 germline knockout mice (R4^KO) to evaluate the effects of acute and endothelial cell-specific Robo4 deletion on HSC trafficking. Strikingly similar to the R4^KO, the acute deletion of Robo4 resulted in altered HSC distribution between the bone marrow and blood compartments, despite normal numbers of VECs and wild-type levels of VCAM1 cell surface protein on sinusoidal VECs. Additionally, consistent with the R4^KO mice, acute loss of Robo4 in the host perturbed long-term engraftment of donor wild-type HSCs and improved HSC mobilization to the peripheral blood. These data demonstrate the significant role that endothelial Robo4 plays in directional HSC trafficking, independent of alterations in VEC numbers and VCAM1 expression.

## Introduction

The bone marrow (BM) vascular endothelium is known both as a physical barrier to HSC trafficking and as a scaffolding for HSC residence. Knopse and colleagues linked the effects of radiation dose on BM vasculature and stroma integrity to improved hematopoietic stem and progenitor cell (HSPC) engraftment through the use of different doses of radiation [1–3]. These data led to additional investigation into the importance and direct role of the vascular barrier for HSC trafficking. Such studies interrogated pathways involved in HSC recovery post-radiation and established a connection to the VCAM1 and integrin α4 pathway [4,5], trafficking mechanisms already well established for leukocytes, but not HSCs. The Wagner and von Andrian groups elegantly revealed the importance of VCAM1 and E- and P-selectins

**Funding:** This work was supported by UCSC and by CIRM Facilities awards CL1-00506 and FA1-00617-1, RRID:SCR_021149, to UCSC. The funders had no role in study design, data collection and analysis, decision to publish, or preparation of the manuscript.

**Competing interests:** The authors have declared that no competing interests exist.

for HSC rolling on the vasculature and subsequent homing to the BM niche [6,7]. This work additionally linked the requirement for VCAM1 and selectins as major pathways for recruitment of HSPCs to the BM after myeloablative radiation preconditioning of recipients of transplanted HSPCs [6,7]. Vascular regulation of HSC trafficking was also associated with the integrin regulation of downstream Rac/Rho pathways and revealed to be instrumental in regulating BM vascular permeability [6,8–10]. After discovering that the endothelial transmembrane protein Robo4 [11–13] is also highly selectively expressed on HSCs [14,15], we pursued its functional role in HSC biology. We uncovered that expression of the transmembrane protein Robo4 by both HSCs [16] and endothelial cells [17] is a key regulator of both HSC location and engraftment potential. We revealed that germline deletion of Robo4 (R4$^{KO}$) resulted in altered HSC location, increased HSC mobilization efficiency in response to the CXCR4 inhibitor AMD3100, increased vascular permeability, decreased numbers of VECs in the BM, and reduced BM sinusoidal VCAM1 expression [16,17]. Intriguingly, reciprocal transplantation experiments revealed perturbed homing and engraftment of R4$^{KO}$ HSCs in Wt mice, and of Wt HSCs in R4$^{KO}$ recipients, demonstrating that both HSC-intrinsic and extrinsic Robo4 expression is essential for optimal HSC function. Here, we sought to address the relationship between germline Robo4 deletion, increased radiation damage, and cellular and molecular changes to the BM architecture upon permanent Robo4 loss. Additionally, we tested whether the acute loss of Robo4, in all cells or only endothelial cells, also altered HSC trafficking, vascular cell phenotypes, HSC mobilization and engraftment, as exhibited by the R4$^{KO}$. Using three Robo4 models–permanent germline deletion, and acute conditional deletion in all cells or selectively in endothelial cells–we provide further insight into the mechanisms involved in HSC trafficking, thereby strengthening the evidence for Robo4 as a key modulator of HSC location.

## Materials and methods

### Mouse lines

All animals were housed and bred in the AAALAC-accredited vivarium at UC Santa Cruz and maintained under guidelines approved by UCSC's animal research ethics committee, the Institutional Animal Care and Use Committee (IACUC). The UCSC IACUC specifically approved this study under protocol ID Forsc2004. Anesthesia for recipients of transplanted hematopoietic cells was administered by isoflurane inhalation. Euthanasia was performed by isoflurane or carbon dioxide inhalation, per approved protocols. The following mice were utilized for these experiments: C57Bl6 (Wt) (JAX, cat#664), BoyJ (JAX, cat#2014), Ubc-GFP (JAX, cat#4353), Robo4 Germline Knockout (R4$^{KO}$, S1A Fig) [18], Robo4$^{lox/lox}$ [19] x B6.129-*Gt(ROSA)26Sor$^{tm1}$* $^{(cre/ERT2)Tyj}$/J (JAX, cat#8463) [20] (referred to as R4$^{cKO}$ post tamoxifen-induced floxing) and the Robo4$^{lox/lox}$ x VeCadherin-CreERT2 (Tg(CDh5-cre/ERT2)1Rha) [21] (referred to as R4$^{ECcKO}$ post tamoxifen-induced floxing, S1B Fig). The R4$^{cKO}$ is a ubiquitous conditional knockout model whereas the R4$^{ECcKO}$ is an endothelial-specific conditional knockout model. Adult mice were used between 8–16 weeks of age and randomized based on sex. Only conditional knockouts and corresponding controls were intraperitoneally (IP) injected with 100 - 150mg/kg tamoxifen (Sigma T5648-1G) resuspended in 5% ethanol and 95% corn oil (Sigma C8267-500ml). Mice were IP injected on days 0, 2, and 4 to ensure high-efficiency floxing of Robo4. Acute model employed either analyzing, transplanting into, or mobilizing floxed mice or treated controls one week post initial tamoxifen injection (3 days post final tamoxifen injection).

### Genotyping primers

Robo4 lox/lox: 314 bp and Wt: 212 bp
 For: `GCTTGTGTCCAGGGAAATACG`

Rev: `TTGGGAAGTCAGCAAATCAGC`
VeCadherin CreERT2: 548 bp and Wt: No Band
 For: `TCC TGA TGG TGC CTA TCC TC`
 Rev: `CCT GTT TTG CAC GTT CAC CG`
Rosa-CreERT2: Cre: 800 bp and Wt: 650bp
 Mutant: `CCT GAT CCT GGC AAT TTC G`
 Common: `AAA GTC GCT CTG AGT TGT TAT`
 Wild Type: `GGA GCG GGA GAA ATG GAT ATG`
Robo4$^{KO}$: Robo4$^{KO}$: 320 bp and Wt: 400 bp
 Primer 1: `AGAACAACCGGACAAAAGTGTATG`
 Primer 2: `GTCTGAGTCCATAGGTCAAGATC`
 Primer 3: `CAGGAAGATGATGAGGTTCCTTGG`
Robo4 floxed/floxed: 400 bp, Wt: 601 bp, R4lox/lox: 903 bp
 For: `CTGTCTGTGGCTGGTGAGG`
 Rev: `TGCTCTTGACACCATCAGGG`

## Mobilization/Tissue isolation

Mice were treated with a single dose of AMD3100 (Sigma A5602) at 5mg/kg subcutaneously (SQ), 1 hour prior to total blood collection [16,17,22–25]. Maximum blood volume was obtained with perfusion by injecting PBS/20 mM EDTA into the left ventricle, clipping the right atrium and collecting the pooled perfused blood in the chest cavity. The total blood was processed for cell counts and flow cytometry analysis, as described previously [16,17,24]. Briefly, cells were pelleted by centrifugation, red blood cells lysed with ACK (hypotonic alkaline saline solution), washed with 2% donor calf serum in PBS to remove EDTA, and incubated with fluorescently conjugated antibodies for flow cytometry analysis.

## Transplantation of whole bone marrow cells

Reconstitution assays were performed by transplanting single cell suspensions of BoyJ (CD45.1) or Ubc-GFP$^+$ whole bone marrow (WBM) from donor mice into recipients preconditioned with a dose of non-myeloablative radiation (260, 525 or 735 rads) or into recipients preconditioned with a myeloablative dose of radiation (1050 rads). Recipients were either non-tamoxifen treated congenic Wt or R4$^{KO}$ mice or tamoxifen treated: controls or R4$^{cKO}$ hosts. Recipient mice were bled at the indicated intervals post-transplantation via the tail vein for analysis of donor chimerism in peripheral blood detectable by GFP or CD45.1 expression. RBC lysed blood was stained with B220-APCy7, CD3-PE or CD3-Alexa Fluor 700, Mac1-PECy7, Ter119-PECy5, Gr1-Pacific Blue, and CD61-APC (Biolegend) to detect mature lineage subtypes for both host and donor mice. Donor bone marrow HSC (Lin$^-$/cKit$^+$/Sca1$^+$/Flk2$^-$/CD45.1$^+$ or Lin$^-$/cKit$^+$/Sca1$^+$/Flk2$^-$/GFP$^+$) populations were stained as previously described [16,17,24–28].

## Flow cytometry

Cell labeling was performed on ice in 1X PBS with 5 mM EDTA and 2% serum. Samples were analyzed for donor chimerism (detectable by CD45.1 or GFP) on an LSRII or AriaIII (Becton Dickinson, San Jose, CA), as described previously [16,17,24,27,29–32]. BM HSC and multipotent progenitor (MPP) populations and BM stromal populations were stained as previously described [16,17]. Briefly, the steady state populations are characterized as peripheral blood (PB) HSCs: Lin$^-$/CD27$^+$/cKit$^+$/Sca1$^+$/Flk2$^-$ [33], PB MPP: Lin$^-$/CD27$^+$/cKit$^+$/Sca1$^+$/Flk2$^+$, BM HSC: Lin$^-$/cKit$^+$/Sca1$^+$/Flk2$^-$/CD34$^-$, BM MPP: Lin$^-$/cKit$^+$/Sca1$^+$/Flk2$^+$/CD34$^+$, Total VEC: CD45$^-$/Ter119$^-$/CD31$^+$/Sca1$^+$, Sinusoidal VEC (SECs): CD45$^-$/Ter119$^-$/CD31$^+$/Sca1$^{mid}$/Tie2$^+$;

MSC: CD45$^-$/Ter119$^-$/CD31$^-$/Sca1$^+$/CD51$^+$ and OBL: CD45$^-$/Ter119$^-$/CD31$^-$/Sca1$^-$/CD51$^+$. All antibodies were purchased from Biolegend, eBioscience, or BD Pharmingen.

## Immunohistochemistry

OCT cryopreservation media and ethanol/dry ice slurry was used to embed Wt and R4$^{KO}$ bones after dissection. Bones were stored at -80°C until sectioned using a cryostat (Leica). A tungsten blade was used to cut bone sections (7–25µm), which were fixed with 4% paraformaldehyde for 15 min at 4°C. Tissue was blocked with 10% goat serum for 1 hour at RT and incubated overnight at 4°C with Rabbit α-Laminin Ab (Sigma, L9393 against LAMA1 (laminin subunit α1)), followed by incubation with conjugated Ab (goat anti-rabbit Alexa 594, Invitrogen) for 1–2 hours at RT and incubated with DAPI for 10 min at RT. Sections were washed with HBSS between each step and cover-slipped with Fluoromount-G prior to imaging with Keyence and Zeiss AxioImager scopes from the UCSC Life Sciences Microscopy Center. Images were analyzed by ImageJ software. Images represent 10X magnification of 7-10-micron thick sections.

## Vascular permeability assays

A modified Miles Assay was utilized to assess *in vivo* vascular permeability [17,34]. R4$^{ECcKO}$ mice were IV injected with Evans Blue (50mg/kg) 3 days post-final injection of tamoxifen. Dye was allowed to leak into tissues for 10 minutes prior to euthanasia by isoflurane inhalation. Vascular leakage was measured as OD650/mass tissue by isolating bones or a section of the small intestine, allowing Evans Blue to leak out of tissues in formamide for 3–5 hours at 55°C, and measuring Evans Blue absorbance of the formamide solution (OD 650).

## qRT-PCR

Quantitative RT-PCR was performed as described previously [14,35], except reactions were conducted on a ABI ViiA7 using the Quantace SensiMixPlus SYBR. Expression of β-actin was used to normalize cDNA amounts between samples.

## Radiation sensitivity assay

Both control and R4$^{KO}$ mice were irradiated (450 rads) using an X-ray tube irradiator (Multirad-160, Precision). Animals were allowed to rest for 24 hours prior to the isolation of bone marrow for HSPC and stromal cell quantification by flow cytometry. Samples were stained and prepared as described above.

## Quantification and statistical analysis

Number of experiments, n, and what n represents can be found in the legend for each figure. Statistical significance was determined by two-tailed unpaired student's T-test or one-way ANOVA followed by Tukey's multiple comparisons test. All data are shown as mean ± standard error of the mean (SEM) representing at least three independent experiments, unless otherwise specified.

# Results and discussion

## The requirement for Robo4-mediated HSC engraftment can be circumvented by increased damage to vascular barriers

Our previous data revealed an impediment in HSC trafficking across Robo4-deficient vascular barriers, as significantly fewer donor HSCs long-term engrafted in germline R4$^{KO}$ mice (**S1A Fig**)

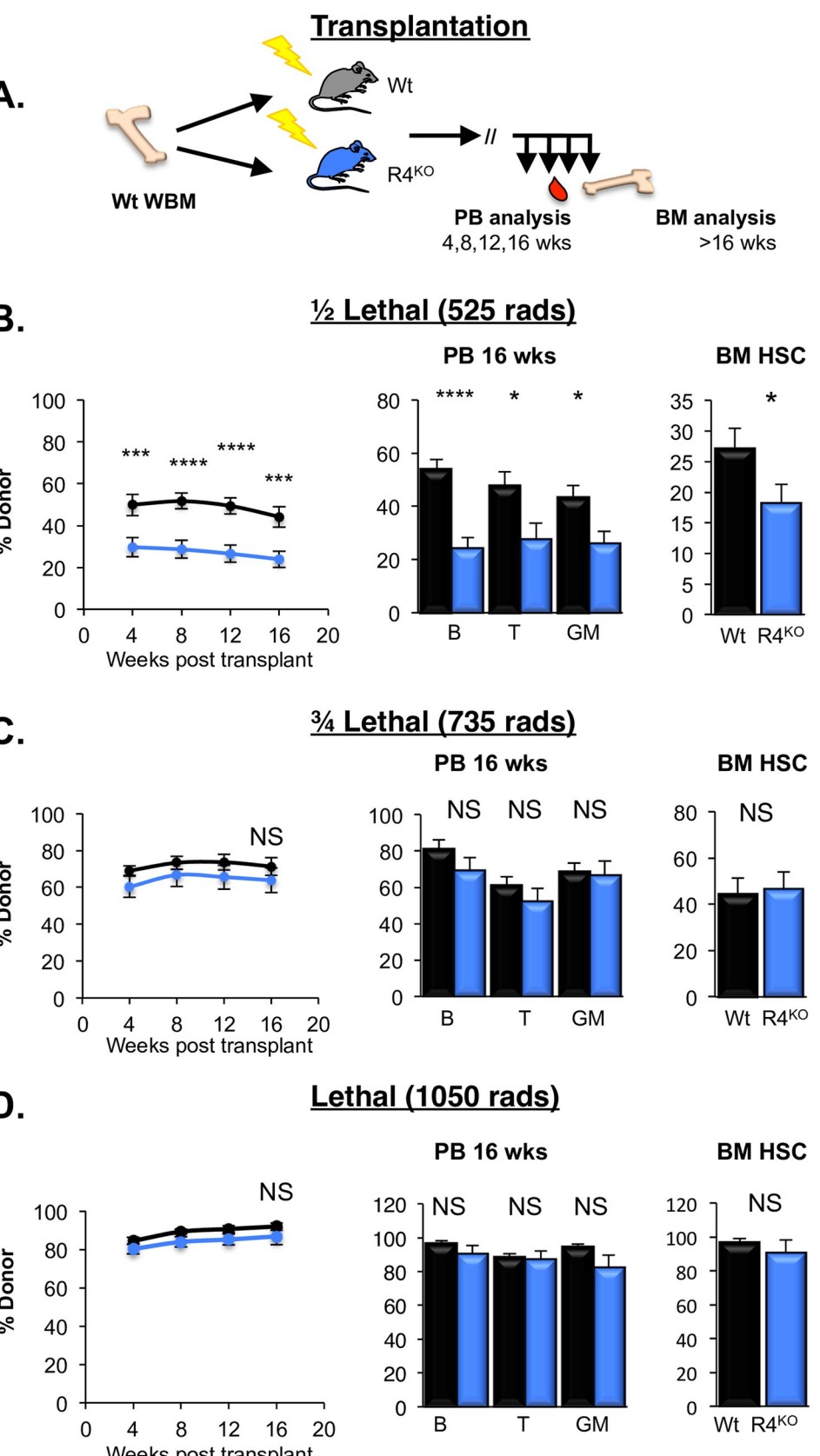

**Fig 1. Requirement for vascular ROBO4 for efficient HSC engraftment can be overcome through increased radiation preconditioning.** A. Schematic of the transplantations into preconditioned recipients at 525, 735, or 1050 rads. B. Loss of Robo4 perturbed HSC reconstitution in R4$^{KO}$ recipients preconditioned with 525 rads compared to Wt recipients. Data represent total donor chimerism in the peripheral blood (PB) over time (line graph), and donor chimerism at the experiment endpoint (>16 wks post-transplantation) of B cells, T cells, and GM cells in the PB and HSCs in the bone marrow (BM). Black bars represent Wt and blue represent R4$^{KO}$. n = 3 independent experiments per radiation dose with n = 21 Wt and n = 21 R4$^{KO}$. We have previously published similar results at this dose [17]; this panel include new data at 525 rads performed side-by-side with the two doses below. Statistics: Unpaired two tailed student's t-test. $^*$ p < 0.05, $^{***}$ p < 0.005 and $^{****}$ p < 0.001. C, D. Increased radiation pre-conditioning resulted in no difference between R4$^{KO}$ and Wt host engraftment potential. Significant engraftment differences between R4$^{KO}$ and Wt recipients was absent with increased radiation pre-conditioning with both ¾ lethal dose (735 rads; **C**) or a lethal preconditioning dose (1050 rads; **D**) resulted in HSC reconstitution in R4$^{KO}$ hosts equivalent to that of Wt hosts. Data panels in **C** and **D** are organized as in panel **B**. Mice in **B** and **C** were transplanted with 7.5M WBM (equivalent to ~500 HSC) and mice in D were transplanted with 1.5M WBM GFP$^+$ cells (~100 HSC). n = 3 independent experiments per radiation dose with n = 14 Wt (C), n = 12 R4KO (C), n = 15 Wt (D) and n = 14 R4$^{KO}$ (D). Statistics: Unpaired two-tailed student's t-test. NS, not significant. See also S1 Fig.

compared to Wt recipients preconditioned with a half-lethal dose of radiation (525 rads; **Fig 1A and 1B**) [17]. Earlier studies had revealed that increased radiation preconditioning improved HSC engraftment [1–3,36]. We therefore hypothesized that increased radiation preconditioning would circumvent the requirement for endothelial Robo4 in promoting HSC engraftment. To test this, we preconditioned Wt and R4$^{KO}$ recipients with increased radiation (735 or 1050 rads; **Fig 1C**). In contrast to low-dose radiation (**Fig 1B**) [17], BM transplantation into these recipients did not result in significant differences in HSC engraftment between Wt and R4$^{KO}$ cohorts, measured by total long-term (>16 weeks) donor reconstitution of mature cells in the PB and of HSCs in the BM (**Fig 1C and 1D**).

The differential engraftment efficiency at low, but not myeloablative, doses of irradiation suggested that Robo4-deficient mice may be more radioresistant, leaving less space for transplanted HSCs to engraft. Reduced numbers of engraftable niches could contribute, together with the previously demonstrated inability to efficiently traverse Robo4-deficient endothelial barriers [17], to poor HSC reconstitution of sublethally irradiated R4$^{KO}$ recipients (**Fig 1B**). We tested this by performing radiation sensitivity experiments, comparing cell numbers in sublethally irradiated WT to R4$^{KO}$ mice (**Fig 2**). We did not detect significant differences in radiosensitivity of WT and R4$^{KO}$ HSPCs, as HSCs, MPPs and myeloid progenitors displayed similar reductions upon irradiation (**Fig 2A and 2C**). In contrast to this observation and to our hypothesis that mice lacking Robo4 would exhibit *less* damage to BM tissue, the radiation experiments demonstrated that R4$^{KO}$ animals had significantly *greater* loss of total BM cells, non-hematopoietic stromal cells, and VECs compared to the WT cohort (**Fig 2B and 2D**). Thus, while the quality of HSC niches was not addressed here, our data clearly show that improved survival of endothelial niche cells does not explain the inefficient HSC engraftment in irradiated Robo4-deficient recipients.

Radiation preconditioning affects the BM sinusoid morphology in mice, which has been linked to improved donor HSC engraftment [1,37]. To determine whether BM sinusoids in Wt and R4$^{KO}$ mice were similarly affected by various doses of radiation preconditioning, we evaluated transverse bone sections post-radiation damage. As we have previously reported [17], there was a significant decrease in total sinusoidal area and average size of individual sinusoids in unconditioned (steady state) R4$^{KO}$ BM compared to Wt BM (**Fig 3A and 3B**). Upon radiation, the BM sinusoids appeared dilated in Wt mice, similar to previous studies [1–3,37]. The BM sinusoids from both Wt and R4$^{KO}$ mice preconditioned with 525 rads exhibited an increase in dilation compared to steady state (**Fig 3A and 3B**). However, those from R4$^{KO}$ mice remained significantly smaller than that of the Wt (**Fig 3A and 3B**). Conversely, myeloablative radiation (1050 rads) led to extensive disruption of sinusoidal structures in both R4$^{KO}$ and Wt mice, erasing the quantitative differences observed at no or low dose irradiation (**Fig 3A and 3B**). Together,

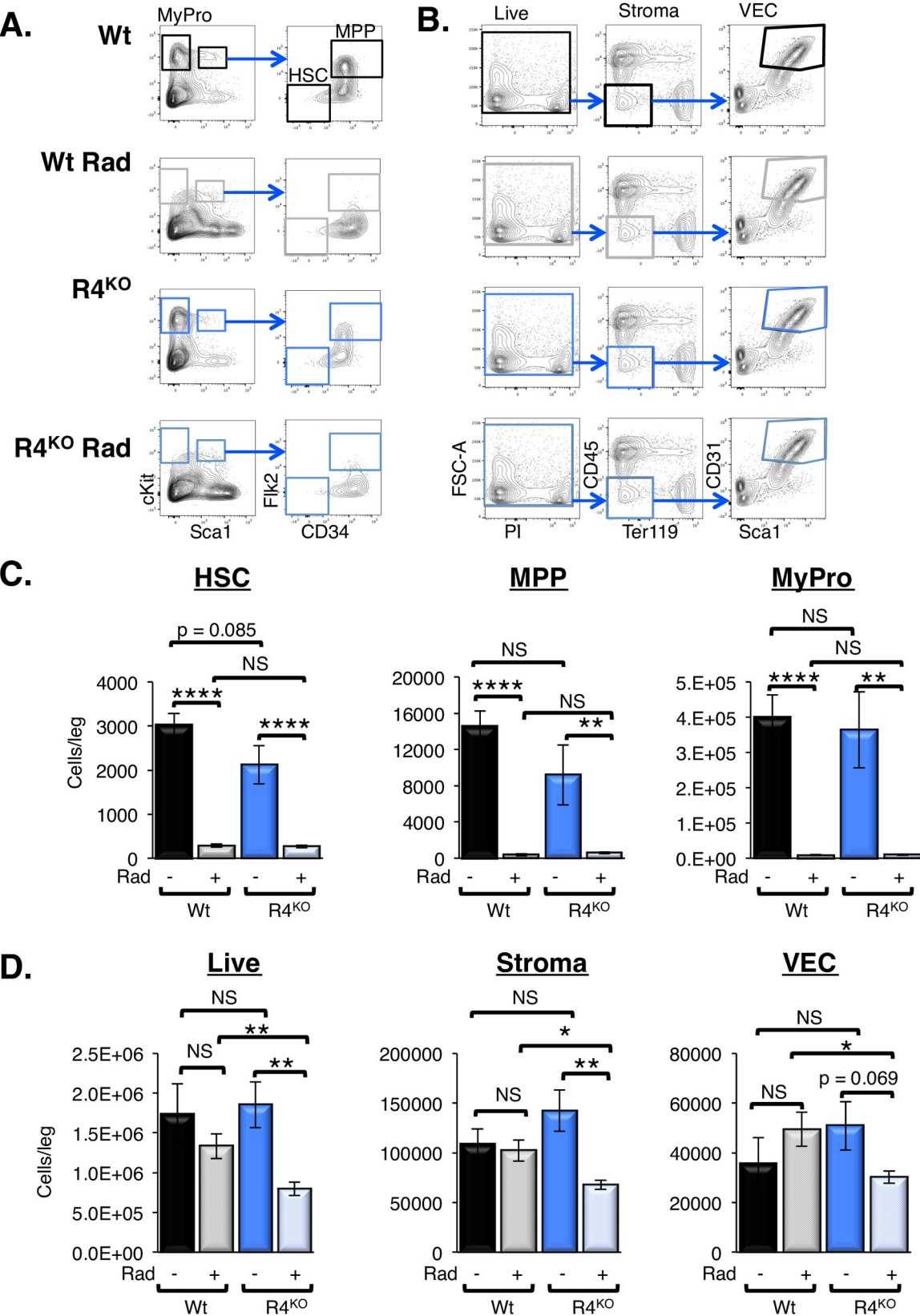

**Fig 2. Hematopoietic stem and progenitors were equally radiation sensitive in R4$^{KO}$ vs Wt mice, but stromal and endothelial cells were more significantly impacted in R4$^{KO}$ mice.** A. Representative flow cytometry plots for the BM HSC, MPP and Myeloid Progenitor (MyPro) populations with and without radiation preconditioning. B. Representative flow cytometry plots for the collagenase-released fraction of the BM for total live, stroma, and VEC (CD45$^-$/Ter119$^-$/CD31$^+$/Sca1$^+$) populations with and without radiation preconditioning. C. Radiation preconditioning significantly impacted the levels of Wt and R4$^{KO}$ HSC, MPP, and MyPro in the bone marrow. However, the levels were not significantly different between Wt or R4$^{KO}$ radiation preconditioned cohorts. Data represent n = 2 independent experiments with n = 6 Wt at 0 rads, n = 8 Wt at 450 rads, n = 6 R4$^{KO}$ at 0 rads, and n = 8 R4$^{KO}$ 450 rads. Statistics: Unpaired two tailed student's t-test. $^*$ p$\leq$ 0.05, $^{**}$ p$\leq$ 0.01, and $^{****}$ p$\leq$0.0001. One-way Anova: HSC p < 0.0001, MPP p < 0.0001, and MyPro p < 0.0001. D. R4$^{KO}$ total live cells, stromal cells, and VECs were significantly more radiation sensitive than the Wt equivalent cell types. Data represent n = 2 independent experiments with n = 6 Wt at 0 rads, n = 8 Wt at 450 rads, n = 6 R4$^{KO}$ at 0 rads, and n = 8 R4$^{KO}$ at 450 rads. Statistics: Unpaired two tailed student's t-test. $^*$ p$\leq$ 0.05, $^{**}$ p$\leq$ 0.01, and $^{****}$ p$\leq$0.0001. One-way Anova: Live p = 0.0049, stroma p = 0.0028, and VEC p = 0.1315.

the engraftment data in R4$^{KO}$ hosts (**Fig 1**) coupled with the sinusoidal VEC (SEC) dilation upon radiation preconditioning (**Fig 3**) suggest that increased radiation damage to sinusoidal structures circumvent the requirement for recipient Robo4 in promoting HSC engraftment.

## Transplanted HSPCs were trapped within the BM vasculature of R4$^{KO}$ mice

Our previous findings, using flow cytometry-based quantification of transplanted HSPCs 3 hours post-transplant, indicated that significantly more HSPCs remained in circulation in R4$^{KO}$ compared to Wt recipients [17]. Concurrently, significantly fewer HSPCs localized to the BM of R4$^{KO}$ compared to Wt recipients [17]. This indicated that donor HSPC extravasation was dependent on recipient Robo4 expression and that in its absence, this process was perturbed. To test this by an alternative method, we transplanted HSPCs into mice preconditioned with a half-lethal dose (525 rads) and evaluated the location of donor HSPCs 3 hours post-transplant in BM sections of R4$^{KO}$ vs Wt hosts by immunofluorescence (**Fig 3C**). These experiments revealed that significantly more (~7 fold) donor GFP$^+$ cells remained intravascular in the R4$^{KO}$ vs Wt recipients, with significantly fewer donor cells localizing to the extravascular BM space in the R4$^{KO}$ hosts (**Fig 3D and 3E**). These data support our previously reported extravasation defects in R4$^{KO}$ mice and reaffirms the importance of vascular Robo4 in HSC trafficking across sinusoidal barriers.

## Acute loss of Robo4 did not alter vasculature or VCAM1 expression

The germline deletion of Robo4 led to changes in both gene and cell surface expression of VCAM1, which is involved in HSC adhesion and cell migration [17]. These findings opened the possibility that VCAM1, rather than Robo4 itself, is directly responsible for the HSC trafficking alterations we observed in R4$^{KO}$ mice. To begin to address this, we determined how acute and endothelial-specific deletion of Robo4 affected HSC location. We employed two tamoxifen-inducible Robo4 conditional knockout mouse models controlled by either Rosa-CreERT2 (ubiquitous) or VeCadherin-CreERT2 (VEC-specific) (**S1B and S1C Fig**) [19–21]. We verified Robo4 deletion using PCR genotyping (**S1Di Fig**) and cell type-selective, significant reductions in Robo4 mRNA levels in sorted VECs and HSCs using qRT-PCR (**S1Dii and S1Diii Fig**). These models allowed us to evaluate the targeted acute loss of Robo4 on both HSC and VEC (R4$^{cKO}$) or selectively on VECs (R4$^{ECcKO}$) compared to R4$^{KO}$ and Wt mice. Utilizing an intraperitoneal injection (IP) protocol (**S1C Fig**), we evaluated the various VEC and HSPC populations 3 days after the final IP injection of tamoxifen. We hypothesized that the acute deletion of Robo4 would not offer a sufficient time interval to alter either the total BM vascular cellularity or the VCAM1 expression on SECs [17], thereby more directly testing the role of Robo4. We first compared the VEC populations from both Robo4 acute deletion models to tamoxifen-treated control mice by flow cytometry (**Fig 4A and 4B**) [17,38,39]. Consistent with

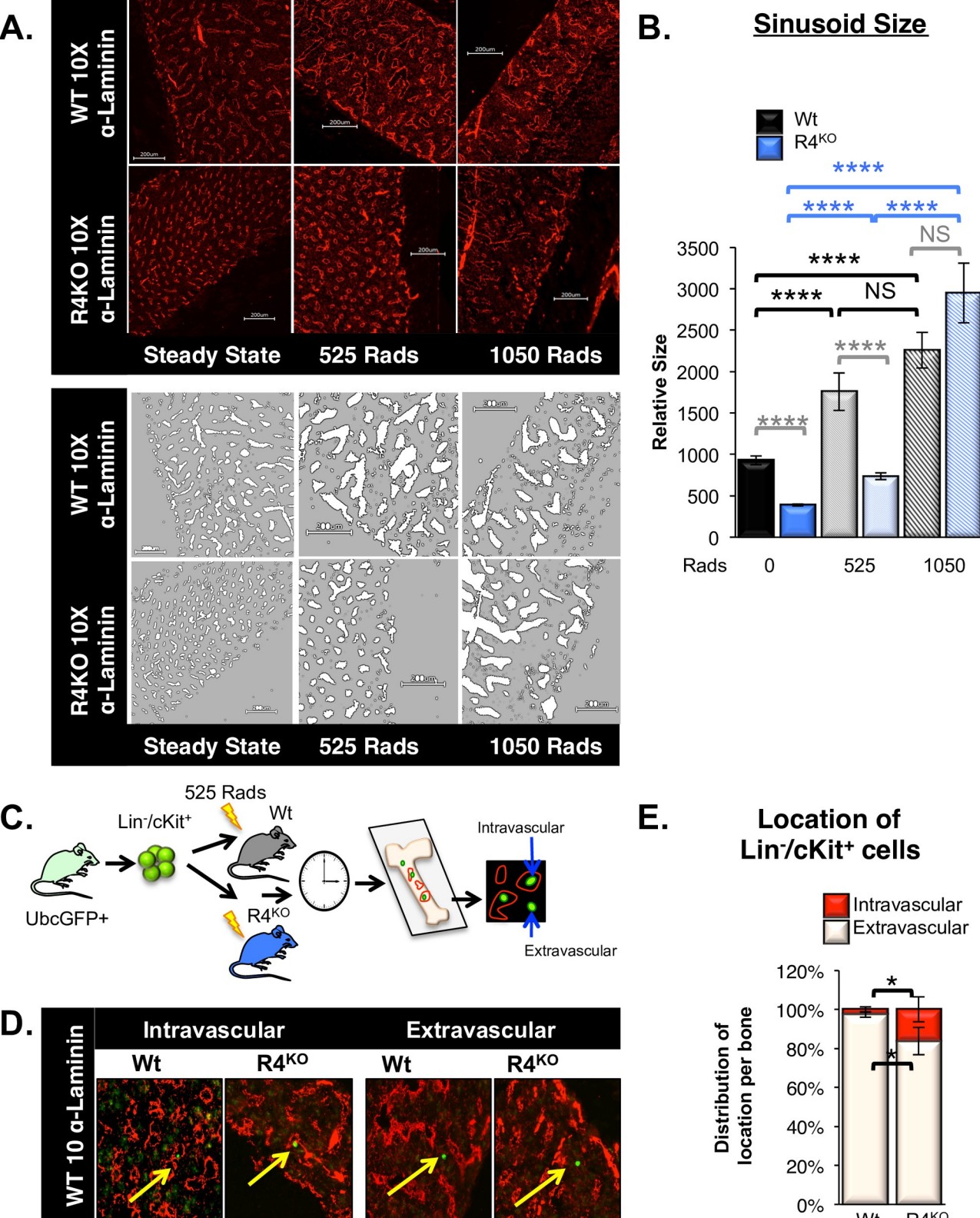

**Fig 3. Increased radiation preconditioning further damages to the vascular bone marrow niche while donor HSC fail to extravasate in R4^KO hosts with non-myeloablative preconditioning.** A. Increased radiation preconditioning caused enlarged BM sinusoidal structures and elimination of the differences between Wt and R4^KO mice. In Wt mouse BM sections, both 525 and 1050 rads resulted in enlarged sinusoidal structures. R4^KO sinusoids

also dilated with increasing radiation, and with 1050 rads the Wt and R4$^{KO}$ mouse sinusoids were enlarged to a similar extent. Representative images depicting BM sections at steady state (no radiation) and 3 hours post sublethal and lethal radiation in both Wt and R4$^{KO}$ mice. The graytone images are binary reconstructions of the sinusoidal structures, using Fiji Software. B. Quantification of the relative size of sinusoidal structures with increased radiation preconditioning. R4$^{KO}$ sinusoids were significantly smaller as compared to Wt at both steady state and 525 rads. R4$^{KO}$ sinusoids significantly dilated with increased radiation and were not significantly different from Wt sinusoids at 1050 rads. Previous comparisons were made with steady state samples. Data represent n = 3 independent experiments with n = 133 Wt sinusoids at 0 rads, n = 108 R4$^{KO}$ sinusoids at 0 rads, n = 89 Wt sinusoids at 525 rads, n = 190 R4$^{KO}$ sinusoids at 525 rads, n = 83 Wt sinusoids at 1050 rads, and n = 90 R4$^{KO}$ sinusoids at 1050 rads. Statistics: Paired two tailed student's t-test. **** p≤0.0001. C. Schematic for short-term homing assay utilizing immunofluorescence to detect the location of transplanted cells in Wt or R4$^{KO}$ BM 3 hours post-transplant. D. Representative images of BM sections depicting the location of transplanted GFP$^+$ HSPCs in Wt and R4$^{KO}$ hosts preconditioned with 525 rads. Donor cells were scored by location as intravascular or extravascular. Images are of 10X magnification. E. Loss of Robo4 resulted in significantly more intravascular HSPCs compared to Wt hosts. Quantification of transplanted GFP$^+$ HSPCs in Wt or R4$^{KO}$ hosts. n = 6 independent experiments, n = 6 mice per strain. Statistics: Paired two tailed student's t-test. * p < 0.05.

our previous report [17], the total numbers of BM VECs increased upon the germline deletion of Robo4 (**Fig 4A and 4B**), whereas the frequency of VCAM1 expression on BM sinusoidal cells decreased (**Figs 3C and 4Ai**). Unlike the germline deletion of Robo4, acute deletion in neither R4$^{cKO}$ nor R4$^{ECcKO}$ mice exhibited significantly elevated VEC numbers or reduced VCAM1 expression on sinusoidal cells 3 days after the final dose of tamoxifen (**Fig 4A–4C**). Similarly, SECs were reduced in the germline R4$^{KO}$ as we have previously reported [17], but not in conditional R4$^{cKO}$ nor R4$^{ECcKO}$, mice (**Fig 4D**). We also evaluated additional BM stromal populations involved in HSC location and engraftment to determine whether Robo4 deletion affected their cellularity, and therefore potentially contributing to defects in HSC engraftment in Robo4-deficient recipients. Neither germline nor acute loss of Robo4 affected either BM mesenchymal stem cell (MSC), osteoblast (OBL), or CD45$^+$ cell numbers (**S2A–S2C Fig**). These data revealed that the vascular alterations evident in the R4$^{KO}$ mice are a consequence of germline deletion and not a resultant mechanism of rapid changes upon acute loss of Robo4.

## Acute loss of Robo4 altered HSC distribution between the BM and blood

We hypothesized that if HSC location is directly dependent on Robo4, HSC levels in the BM and blood would be significantly altered in the acute R4 knockout models, despite their lack of vascular and VCAM1 alterations. Remarkably, the acute loss of Robo4 alone was sufficient to significantly alter HSC levels in both the BM and blood. Significantly more HSCs were present in the blood in both acute conditional Robo4 deletion models (**Fig 5A and 5B**), with a reciprocal decrease of BM HSCs as compared to tamoxifen-treated control mice (**Fig 5C and 5D**). These data were similar to our previous R4$^{KO}$ findings (**Fig 5B and 5D**) [17]. MPP cellularity levels were unchanged in both the PB and the BM in the R4$^{KO}$ [16,17,33], as well as in the two conditional models (**S3A and S3B Fig**). These data strengthen our previous finding that Robo4 is an important and specific regulator of HSC location.

## Vascular permeability increased with acute loss of Robo4

We and others have shown that the germline loss of Robo4 results in an increase in vascular permeability [17,18,40]. Our previous data revealed that despite this leakiness, engraftment was *decreased* in R4$^{KO}$ mice [17]. To test whether acute loss of vascular Robo4 is sufficient to increase vascular permeability, we employed the Miles permeability assay [17,18,34,40] in R4$^{ECcKO}$ mice. The results revealed that acute, endothelial-selective loss of vascular Robo4 was sufficient to induce significant leakiness (**Fig 6A**), similar to earlier reported findings upon Robo4 germline deletion [17,18,34,40]. These data show that alterations in VEC numbers or VCAM1 expression are not integral for the role that Robo4 plays in vascular stability.

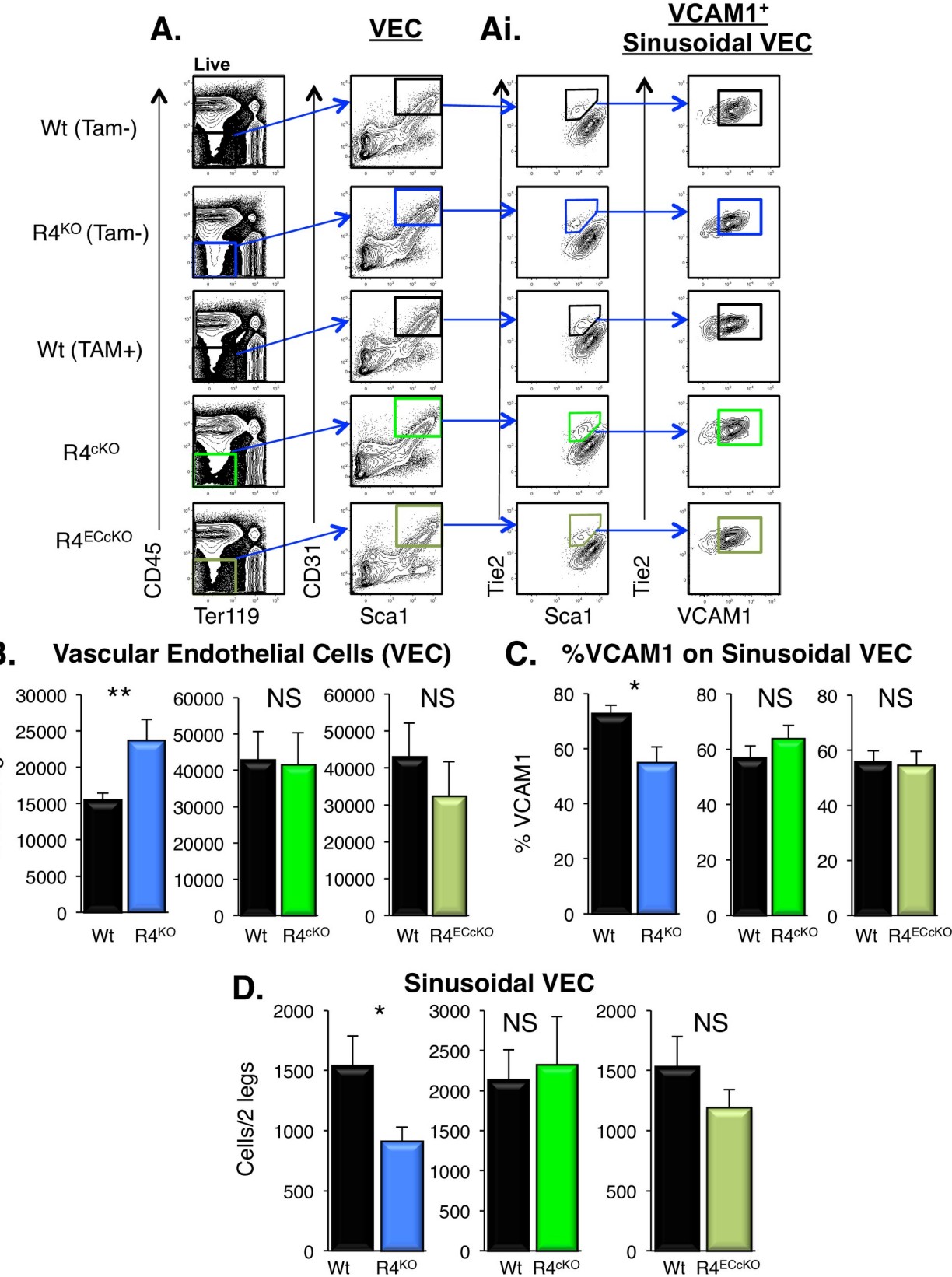

**B.** Vascular Endothelial Cells (VEC)

**C.** %VCAM1 on Sinusoidal VEC

**D.** Sinusoidal VEC

**Fig 4. Acute deletion of Robo4 did not alter BM VECs or sinusoidal VCAM1 expression.** A. Representative flow cytometry plots for the BM VEC and SEC populations. VECs were defined as CD45$^-$/Ter119$^-$/CD31$^+$/Sca1$^+$ BM cells. Ai. SECs were additionally defined as Tie2$^+$ and Sca$^{mid}$ and evaluated for VCAM1 expression. B. Germline, but not acute, deletion of Robo4 resulted in increased numbers of BM VECs. The germline R4$^{KO}$ mice exhibited significantly more VECs compared to Wt mice, whereas neither the pan- (R4$^{cKO}$ by Rosa26-CreERT2) nor vascular-specific (R4$^{ECcKO}$ by VEcad-CreERT2) acute Robo4 deletion exhibited significantly different numbers of total VECs. We have previously published similar results for the R4$^{KO}$ mice [17]; this panel includes both previously reported and new data from the R4$^{KO}$ performed side-by-side with the two conditional models. Data represent n = 4–6 independent experiments with n = 37 wt, n = 37 R4$^{KO}$, n = 18 tamoxifen treated Wt, n = 15 R4$^{cKO}$, and n = 10 R4$^{ECcKO}$. Statistics by unpaired two tailed student's t-test. ** p < 0.01. C. Unlike the germline R4$^{KO}$, the frequency of VCAM1 on sinusoidal BM VECs was not significantly different upon acute pan or endothelial-specific deletion of Robo4. R4$^{KO}$ mice exhibited significantly lower frequency of VCAM1$^+$ sinusoidal VECs compared to Wt cohorts, whereas R4$^{cKO}$ or R4$^{ECcKO}$ did not. We have previously published similar results for the R4$^{KO}$ mice; this panel includes both previously reported and new data from the R4$^{KO}$ performed side-by-side with the two conditional models [17]. Data represent n = 4 independent experiments, n = 21 Wt, n = 21 R4$^{KO}$, n = 18 tamoxifen treated Wt, n = 15 R4$^{cKO}$, and n = 10 R4$^{ECcKO}$. Statistics by unpaired two tailed student's t-test. * p < 0.05. See also **S2 Fig**. E. Unlike the germline R4$^{KO}$, the number of BM SECs (CD45$^-$/Ter119$^-$/Sca1$^{low}$/Tie2$^+$) was not significantly different upon acute pan or endothelial-specific deletion of Robo4. R4$^{KO}$ mice exhibited significantly lower BM sinusoidal VECs compared to Wt cohorts, whereas R4$^{cKO}$ or R4$^{ECcKO}$ did not. We have previously published similar results for the R4$^{KO}$ mice; this panel includes both previously reported [17] and new data from the R4$^{KO}$ performed side-by-side with the two conditional models. Data represent n = 4 independent experiments, n = 21 Wt, n = 21 R4$^{KO}$, n = 18 tamoxifen treated Wt, n = 15 R4$^{cKO}$, and n = 10 R4$^{ECcKO}$. Statistics by unpaired two tailed student's t-test. * p < 0.05.

## Acute loss of Robo4 significantly improved AMD3100-mediated HSC mobilization

Previously, we showed that Robo4 deletion significantly improved AMD3100-mediated HSC mobilization compared to Wt cohorts [16,17]. We hypothesized that the acute loss of Robo4 ubiquitously or on VECs alone would be sufficient to improve HSC mobilization in the absence of vascular differences as compared to Wt controls [16,22] (**Fig 6B**). Indeed, similar to the R4$^{KO}$ mice [16,17], both Robo4 conditional deletion models exhibited significantly improved AMD3100-mediated mobilization; ~4000 and ~3000 phenotypic HSCs mobilized in the R4$^{cKO}$ and R4$^{ECcKO}$ models respectively, compared to ~1200 phenotypic HSC in the tamoxifen treated controls (**Fig 6C**). These data strengthen our previous, unexpected finding that loss of Robo4 facilitates BM-to-blood trafficking (**Fig 6C**), while HSC movement in the opposite direction is impaired upon Robo4 deletion (**Figs 1B, 3D and 3E**) [16,17].

## Long-term engraftment was perturbed in Robo4 conditional knockout recipients

In germline R4$^{KO}$ recipients, both short-term homing [17] and long-term engraftment were impaired (**Fig 1B**) [17]. Mimicking the R4$^{KO}$ transplantation models, we tested whether the acute deletion of Robo4 was sufficient to perturb long-term HSC engraftment in recipients preconditioned with 525 rads (**S4A Fig**). Surprisingly, there were no significant differences in donor-derived total or mature cell subsets between the R4$^{cKO}$ and tamoxifen-treated Wt controls (**S4B and S4C Fig**). Previous reports indicated that tamoxifen affects both vascular blood flow and apoptosis of resident HSPCs [41,42]. We speculated that, similar to elevated doses of radiation in the R4$^{KO}$ (**Fig 1C and 1D**), the effects of tamoxifen treatment, when coupled with 525 rad preconditioning, could explain the lack of differences in engraftment (**S4B and S4C Fig**). We therefore tested whether HSC engraftment was impaired in the R4$^{cKO}$ vs tamoxifen-treated controls if recipients were preconditioned with a lower dose of radiation. We transplanted Wt BM cells into tamoxifen-treated R4$^{cKO}$ and Wt recipients preconditioned with only a 260 rad dose and evaluated long-term engraftment. Intriguingly, and consistent with our previous findings (**Fig 1B**), recipient mice lacking Robo4 at time of transplant failed to efficiently engraft compared to tamoxifen-treated control mice (**Fig 6D–6F**) [17]. Total donor chimerism was significantly lower in the peripheral blood over 16 weeks post-transplant in the R4$^{cKO}$ compared to controls (**Fig 6E**). There was a ubiquitous reduction in donor-derived cells

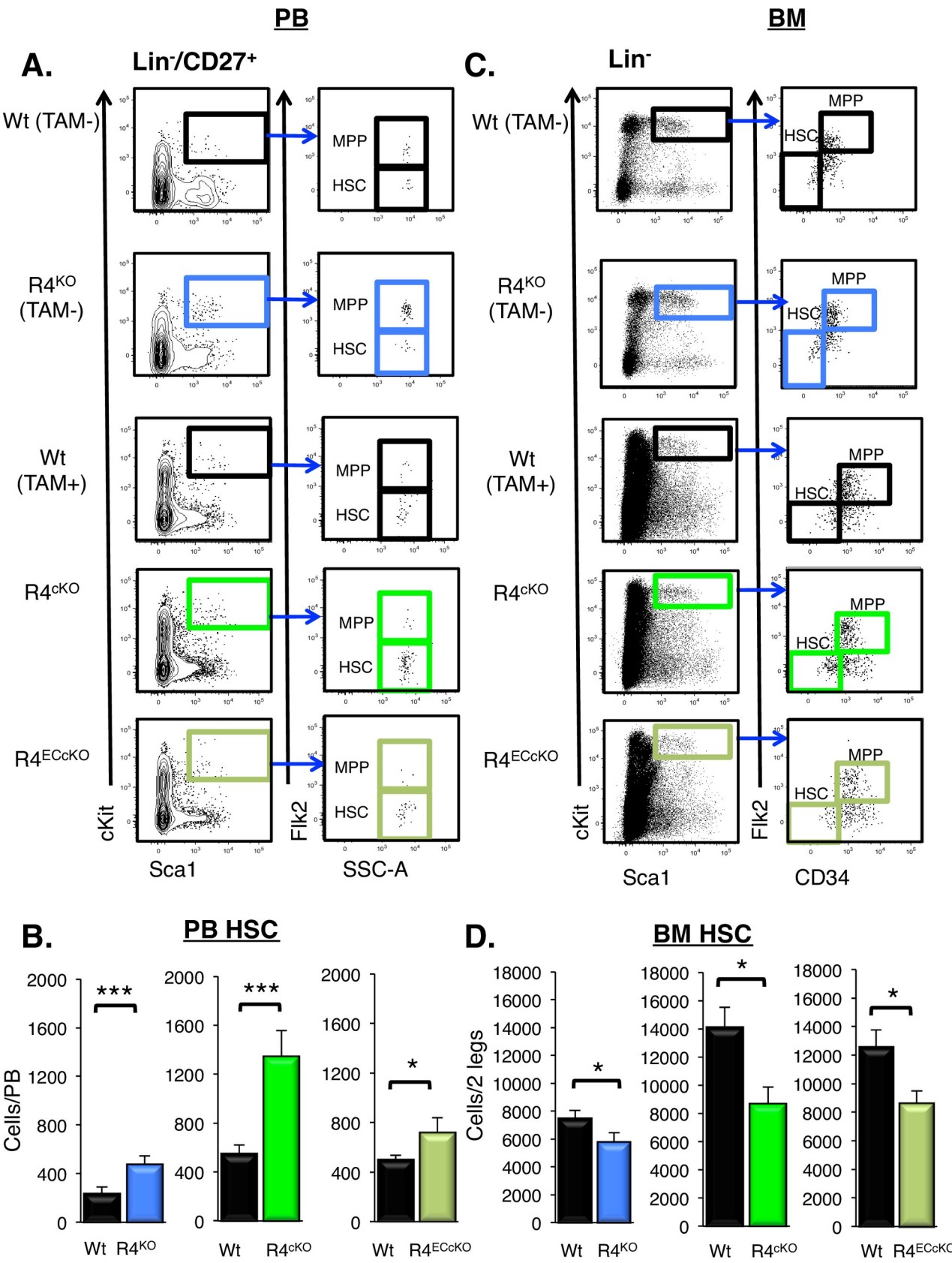

**Fig 5. Acute pan- or endothelial cell-specific deletion of Robo4 was sufficient to alter the location of HSCs.** A. Representative flow cytometry plots of peripheral blood HSCs and multipotent progenitor cells (MPPs). Similar to that of germline R4$^{KO}$ mice, the numbers of HSCs were significantly elevated in the blood of R4$^{cKO}$ and R4$^{ECcKO}$ mice. We have previously published similar results for the R4$^{KO}$ mice; this panel includes both previously reported and new data from the R4$^{KO}$ performed side-by-side with the two conditional models [16]. Data represent n = 5–9 independent experiments, with n = wt, n = R4$^{KO}$, n = 33 Tamoxifen treated Wt, n = 29 R4$^{cKO}$, and n = 18 R4$^{ECcKO}$ mice. Statistics by unpaired two tailed student's t-test * p≤0.05, *** p≤0.001. B. Representative flow cytometry plots of BM HSCs and MPPs. C. Significantly fewer HSCs in the BM of R4$^{KO}$, R4$^{cKO}$ and the R4$^{ECcKO}$ mice as compared to the respective Wt control mice. We have previously published similar results for the R4$^{KO}$ mice; this panel includes both previously reported [16] and new data from the R4$^{KO}$ mice performed side-by-side with the two conditional models. Data represent n = 5–9 independent experiments, with n = 25 wt, n = 9 R4$^{KO}$, n = 12 Tamoxifen treated Wt, n = 16 R4$^{cKO}$, and n = 10 R4$^{ECcKO}$ mice. Statistics by unpaired two tailed student's t-test * p≤0.05, *** p≤0.001. See also **S3 Fig**.

across the predominant mature lineages: B220$^+$ B cells, CD3$^+$ T cells, Mac1$^+$/Gr1$^+$ myeloid cells, CD61$^+$ platelets, and Ter119$^+$ red blood cells (**Fig 6F**). These data strengthen Robo4 as a key player in HSC trafficking and BM engraftment at time of transplant.

We have established that VCAM1 expression was downregulated on R4$^{KO}$ SECs, but acute deletion of Robo4 was not sufficient to immediately (within 3 days) contribute to VCAM1 changes at steady state (**Fig 4A and 4C**). We hypothesized that, similar to reported findings [43], sinusoidal VCAM1 would remain at Wt levels immediately post-radiation in the R4$^{cKO}$ mice. To test this, we evaluated the VCAM1 expression on SECs immediately post-radiation, at time of transplant. Remarkably, VCAM1 cell surface expression was not affected in R4$^{cKO}$ compared to Wt controls (**Fig 6G**). We next investigated whether VCAM1 expression changed over time post-radiation preconditioning and found that VCAM1 was significantly downregulated 16 weeks post radiation compared to Wt recipients (**Fig 6G**). These data thus revealed that the conditional loss of Robo4 was sufficient to alter VCAM1 expression over time, similar to that of R4$^{KO}$ mice. Collectively, these findings showed that reduced HSC homing and engraftment in Robo4-deficient BM cannot be attributed to reduced levels of VCAM1, because its downregulation upon Robo4 deletion is not apparent until after an extended time period. Importantly, the poor donor HSC reconstitution (**Fig 6E**) occurred despite normal levels of sinusoidal VCAM1 expression (**Fig 6G**) in the R4$^{cKO}$ mice at time of transplant. This "acute" VCAM1 independence does not preclude a role for VCAM1 in the longer-term effects of Robo4 deletion.

## Vascular Robo4 regulates HSC trafficking independent of VCAM1 downregulation

Collectively, our new results solidify the importance of Robo4 as a key regulator of HSC trafficking to and from the bone marrow niche. We confirmed that Robo4 is an important facilitator of HSC extravasation from the vasculature, unless sinusoidal structures were severely damaged by preconditioning regimens (**Figs 1–3**). Using conditional Robo4 deletion mouse models, we evaluated whether HSC trafficking was modulated by acute deletion of Robo4 or as a consequence of other key regulators mis-regulated in the germline R4$^{KO}$ mice. Specifically, we discoupled the role of Robo4 from the cell surface receptor VCAM1, a key regulator of HSPC adhesion, trafficking and homing into the bone marrow [5,6,36,44,45]. Germline, but not acute, loss of Robo4 perturbed sinusoidal VCAM1 expression, a key regulator in HSC trafficking. Acute loss of Robo4, on either all cells or only endothelium, did not affect either VEC numbers or VCAM1 expression on sinusoidal cells that are involved in HSC trafficking (**Fig 4**). Still, acute Robo4 deletion was sufficient to alter the location of HSCs at steady state, driving significantly more HSCs into the periphery (**Fig 5**). The VeCadherin deletion model clearly demonstrated that endothelial Robo4, in addition to hematopoietic Robo4 [16], is an important regulator of HSC location. Similar to germline deletion, acute loss of Robo4 also induced vascular leak (**Fig 6A**) and improved HSC mobilization (**Fig 6C**). We showed that VCAM1

**A. Permeability Assay**

**B.**

**C. Flk2- HSC**

**D.**

**E. ¼ Lethal (260 rads)**

**F. ¼ Lethal (260 rads)**

**G. Fold VCAM1 post rad**

**Fig 6. Acute loss of Robo4 resulted in increased vascular permeability, improved HSC mobilization, and perturbed long-term engraftment.** A. Acute loss of vascular Robo4 resulted in increased vascular permeability in the small intestine. n = 4 independent experiments with, n = 11 tamoxifen treated Wt and n = 13 R4$^{ECcKO}$. Statistics by unpaired two-tailed student's t-test. $^*$ p < 0.05. B. Schematic depicting tamoxifen treatment of the Robo4 conditional knockout mice prior to AMD3100 mobilization. C. Improved AMD3100-mediated HSC mobilization upon both germline and conditional R4 deletion compared to Wt cohorts. n = 3–5 experiments, with n = 16 tamoxifen treated Wt without AMD3100, n = 19 tamoxifen treated Wt with AMD3100, n = 10 R4$^{cKO}$ without AMD3100, n = 14 R4$^{cKO}$ with AMD3100, n = 4 R4$^{ECcKO}$ without AMD3100 and n = 5 R4$^{ECcKO}$ with AMD3100. Statistics: One-way Anova with Tukey's multiple comparison test (black) and unpaired two-tailed student's t-test (blue) $^*$ p≤0.05, $^{**}$ p≤0.005, $^{***}$ p≤0.001, $^{****}$ p≤0.0001. D. Schematic depicting the tamoxifen injections to delete Robo4, followed by radiation treatment (3 days post final tamoxifen dose), and analysis at day 1 of radiation or 16 weeks post-transplant of VCAM1 expression on sinusoidal VECs. E. R4$^{cKO}$ mice preconditioned with ¼ lethal radiation (260 rads) exhibited significantly lower engraftment of donor HSCs compared to the tamoxifen-treated control hosts over 16 weeks post-transplant. 10M GFP$^+$ WBM cells (667 HSC equivalent) were transplanted per recipient. n = 3 experiments, with n = 10 tamoxifen treated Wt and n = 13 R4$^{cKO}$. Statistics by unpaired two tailed student's t-test. $^*$ p≤0.05, $^{**}$ p≤0.005. F. Peripheral blood donor chimerism of mature hematopoietic lineages >16 wks post-transplantation. Donor chimerism of all leukocytes, RBCs and platelets (plt) was lower in the R4$^{cKO}$ hosts compared to Wt controls; cohorts are the same as in D. Statistics by unpaired two tailed student's t-test. $^*$ p≤0.05, $^{***}$ p≤0.001. G. VCAM1 surface protein is reduced over time post-acute loss Robo4 and radiation exposure. At day 1 of radiation (day of HSC transplant), VCAM1 cell surface expression was equivalent in R4$^{cKO}$ hosts compared to control Wt host mice. However, at 16 weeks post-transplant, VCAM1 was significantly downregulated on sinusoidal VECs in the R4$^{cKO}$ hosts, similar to that of the germline R4$^{KO}$ mice. Day 1: n = 3 experiments, n = 3–5 mice per cohort; week 16: n = 3 tamoxifen treated Wt and n = 4 R4$^{cKO}$. Statistics: Unpaired two tailed student's t-test $^*$ p≤0.05, $^{**}$ p≤0.005, $^{***}$ p≤0.001. See also S4 Fig.

was not downregulated in the acute deletion models, even with radiation preconditioning (**Fig 6G**), and thus was not responsible for the poor HSC engraftment in the R4$^{cKO}$ compared to Wt controls (**Fig 6E and 6F**). VCAM1 was, however, downregulated over time in the R4$^{cKO}$ mice (**Fig 6G**), demonstrating that Robo4 loss has long-term consequences that were not readily detected soon after deletion. Collectively, these data strengthen the role of Robo4 as critical player in directional HSC trafficking independently of VCAM1 and provides more insight into the complex molecular mechanisms that regulate VEC and HSC trafficking. Further understanding how vascular integrity modulates HSC trafficking could improve both HSC mobilization and donor engraftment in the clinic, potentially by repurposing already approved vascular modulators, such as Viagra [25].

## Supporting information

**S1 Fig. Robo4 deletion models.** A. Robo4 germline knockout model [18,40]. Alkaline phosphatase was inserted between exons 2–4. B. Schematic of tamoxifen injection schedule. Mice (**Figs 4–6**) were analyzed on day 7, 3 days after final IP injection of tamoxifen. C. Robo4 conditional knockout model [19], RosaCreERT2 [20], and VeCadherin-CreERT2 [21]. LoxP sites were inserted flanking exon 3 of the Robo4 gene. D. Genetic deletion in the Robo4 locus and reduced Robo4 mRNA levels in the three mouse models. Di. Image depicted shows PCR-based banding patterns for DNA isolated from ear clips of R4$^{lox/lox}$, Wt, and R4$^{cKO}$ (in duplicate) mice, VECs sorted from Wt tamoxifen-treated mice (in triplicate), R4$^{cKO}$ tamoxifen-treated mice (in triplicate), and R4$^{ECcKO}$ tamoxifen-treated mice (in triplicate). The estimated banding patterns: untreated R4$^{lox/lox}$: 903bp, Wt: 601 bp, and tamoxifen-treated R4$^{cKO}$ and R4$^{ECcKO}$ (floxed exon deleted): 400bp. (Relative gene expression of mRobo4. (ii) The VEC and (iii) HSC populations were isolated by FACS from untreated (no tamoxifen) Wt or R4$^{KO}$ mice or from Wt, R4$^{cKO}$, and R4$^{ECcKO}$ mice post tamoxifen treatment (treatment schedule as in panel C). Robo4 mRNA levels were normalized to β-actin levels. Robo4 mRNA levels were reduced in VECs in all 3 models, as expected (panel Dii). In HSCs, Robo4 mRNA levels were reduced in R4$^{KO}$ and R4$^{cKO}$, but not in R4$^{ECcKO}$, mice, also as expected (panel Diii). Data represent n = 3 experiments from 2–3 mice per cohort per sort. Statistics by unpaired two tailed student's t-test $^{**}$ p < 0.01 and $^{****}$ p < 0.0001.
(PDF)

**S2 Fig. Mesenchymal and osteoblast stromal cell numbers were not perturbed with loss of Robo4.** A. Germline, acute, or endothelial-specific deletion of Robo4 did not result in significantly altered cell numbers of bone marrow mesenchymal stem cells (MSCs). Data represent n = 4–6 independent experiments with n = 37 Wt, n = 37 R4$^{KO}$, n = 18 tamoxifen treated Wt, n = 15 R4$^{cKO}$, and n = 10 R4$^{ECcKO}$. Statistics by unpaired two tailed student's t-test showing no significance. B. Germline, acute, or endothelial-specific deletion of Robo4 did not result in significantly altered cell numbers of bone marrow osteoblasts (OBL). Data represent n = 4–6 independent experiments with n = 37 Wt, n = 37 R4$^{KO}$, n = 18 tamoxifen treated Wt, n = 15 R4$^{cKO}$, and n = 10 R4$^{ECcKO}$. Statistics by unpaired two tailed student's t-test, showing no significance. C. Germline, acute, or endothelial-specific deletion of Robo4 did not result in significantly altered cell numbers of bone marrow CD45+ hematopoietic cells. Data represent n = 4–6 independent experiments with n = 10 Wt, n = 10 R4$^{KO}$, n = 12 tamoxifen treated Wt, n = 12 R4$^{cKO}$, and n = 14 R4$^{ECcKO}$. Statistics by unpaired two tailed student's t-test, showing no significance.
(PDF)

**S3 Fig. Robo4 deletion does not alter location of MPP.** A. Unlike HSCs, MPPs were not significantly increased in the blood of Robo4-deficient mice. We have previously published similar results for the R4$^{KO}$ mice; this panel includes both previously reported and new data from the R4$^{KO}$ performed side-by-side with the two conditional models [16]. Data represent n = 5–9 independent experiments, with n = Wt, n = R4$^{KO}$, n = 33 Tamoxifen treated Wt, n = 29 R4$^{cKO}$, and n = 18 R4$^{ECcKO}$ mice. Statistics by unpaired two tailed student's t-test, showing no significance. B. Unlike HSCs, MPPs were not significantly decreased in the BM of Robo4-deficient mice. We have previously published similar results for the R4$^{KO}$ mice; this panel includes previously reported data from the R4$^{KO}$ performed side-by-side with the two conditional models [16]. Data represent n = 5–9 independent experiments, with n = 25 Wt, n = 9 R4$^{KO}$, n = 12 Tamoxifen treated Wt, n = 16 R4$^{cKO}$, and n = 10 R4$^{ECcKO}$ mice. Statistics by unpaired two tailed student's t-test, showing no significance.
(PDF)

**S4 Fig. Increased radiation damage eliminated the differences in engraftment efficiency of donor HSPCs in R4$^{cKO}$ mice.** A. Schematic of tamoxifen injection schedule and radiation/transplantation 3 days post final tamoxifen injection. B. Equivalent peripheral blood donor chimerism levels in R4$^{cKO}$ hosts compared to Wt hosts treated as in A. Statistics by unpaired two tailed student's t-test, showing no significance. C. No difference in donor lineages in the transplanted mice from B between R4$^{cKO}$ or controls. Recipients were transplanted with 7.5 M UbcGFP+ WBM cells (500 HSC equivalent). n = 3 independent experiments per radiation dose with n = 8 tamoxifen treated Wt and n = 11 R4$^{cKO}$. Statistics by unpaired two tailed student's t-test, showing no significance.
(PDF)

## Acknowledgments

We thank Smrithi Rajendiran, Bryce Manso, and Alessandra Rodriguez y Baena for manuscript review and Christina De Leon, Mark Landon, and Tuan Vo for animal care. We thank Dr. Ben Abrams and Bari Holmes Nazario at UCSC's Institute for the Biology of Stem Cells (IBSC) Microscopy and Flow Cytometry Core Facilities, respectively, for assistance with imaging and flow cytometry. We thank Drs. Dean Li for the Robo4 germline knockout (R4$^{KO}$) mice, Xiaobing Yuan for the Robo4$^{lox/lox}$ mice, and Eugene Butcher the VeCadherin-CreERT2 mice with permission by Dr. Ralph Adams.

## Author Contributions

**Conceptualization:** Stephanie Smith-Berdan, Lindsay Hinck, E. Camilla Forsberg.

**Data curation:** Stephanie Smith-Berdan.

**Formal analysis:** Stephanie Smith-Berdan, E. Camilla Forsberg.

**Funding acquisition:** Lindsay Hinck, E. Camilla Forsberg.

**Investigation:** Stephanie Smith-Berdan, Alyssa Bercasio, Leah Kramer, Bryan Petkus.

**Methodology:** Stephanie Smith-Berdan, Lindsay Hinck.

**Supervision:** Stephanie Smith-Berdan, Lindsay Hinck, E. Camilla Forsberg.

**Visualization:** Stephanie Smith-Berdan.

**Writing – original draft:** Stephanie Smith-Berdan.

**Writing – review & editing:** Stephanie Smith-Berdan, Alyssa Bercasio, Leah Kramer, Bryan Petkus, Lindsay Hinck, E. Camilla Forsberg.

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
