## [Decision Letter · Decision Letter 0]

5 Mar 2021

PONE-D-21-02612

Acute and endothelial-specific Robo4 deletion affect hematopoietic stem cell trafficking independent of VCAM1

PLOS ONE

Dear Dr. Forsberg,

Thank you for submitting your manuscript to PLOS ONE. After careful consideration, we feel that it has merit but does not fully meet PLOS ONE’s publication criteria as it currently stands. Therefore, we invite you to submit a revised version of the manuscript that addresses the points raised during the review process.

Overall, the 2 reviewers believed the findings would be of significant interest to the field and had only minor concerns.  There were a few suggestions for providing additional data or discussion to improve the experimental rigor.  In particular, deletion of Robo4 in ECs is one point that needs attention.  Additionally, please address the other points either with new data or by discussion as required.

We look forward to receiving your revised manuscript.

Kind regards,

Kevin D Bunting

Academic Editor

PLOS ONE

Journal Requirements:

"This work was supported by

468 UCSC and by CIRM Facilities awards CL1-00506 and FA1-00617-1 to UCSC. The funders had no role

469 in study design, data collection and analysis, decision to publish, or preparation of the manuscript."

 "No"

"No."

4. Please amend the manuscript submission data (via Edit Submission) to include authors Alyssa Bercasio, Leah Kramer, Bryan Petkus, and Lindsay Hinck

5. As part of your revisions please include the following details: (1) monitoring parameters of your animals, the frequency with which they were checked for health and well-being, and details of humane endpoints; (2) rate of mortality; (3) unanticipated adverse events that took place during the animal study and how you managed/resolved them. We thank you for your cooperation in this matter.

Reviewers' comments:

Reviewer's Responses to Questions

**Comments to the Author**

1. Is the manuscript technically sound, and do the data support the conclusions?

Reviewer #1: Yes

Reviewer #2: Yes

2. Has the statistical analysis been performed appropriately and rigorously? 

Reviewer #1: Yes

Reviewer #2: Yes

3. Have the authors made all data underlying the findings in their manuscript fully available?

Reviewer #1: Yes

Reviewer #2: Yes

4. Is the manuscript presented in an intelligible fashion and written in standard English?

Reviewer #1: Yes

Reviewer #2: Yes

5. Review Comments to the Author

Reviewer #1: This manuscript extends the authors previous findings that Robo4 is extrinsically required for normal HSC trafficking. First, the authors confirm previous studies demonstrating that WT BMCs show decreased donor repopulation in Robo4KO recipients that receive a sublethal dose of radiation (535 Rads) compared to WT recipients, and demonstrate no difference in the repopulation of WT BMCs in Robo4KO recipients that received high doses of radiation (725 and 1050 Rads). The authors demonstrate that Robo4KO mice show reduced sinusoidal area in BM sections stained with laminin compared to WT mice. Furthermore, while sinusoidal area is increased after irradiation, the sinusoidal area in Robo4KO mice is reduced compared to controls after 525 Rads radiation. However, while a difference in sinusoidal area is observed in mice that received 525 Rads, no difference in sinusoidal area is observed with increasing doses of irradiation (725 and 1050 Rads). The authors provide additional evidence for defective homing and extravasation of WT BMCs in Robo4KO mice in laminin-stained BM sections. The authors demonstrate that acute (3 day) loss of Robo4 does not affect total VEC cellularity, and VCAM1 expression on SECs, but does affect HSC mobilization in Rob4cKO and Robo4ECcKO mice. The authors provide data that acute loss of Robo4 in Robo4ECcKO mice results in increased vascular permeability. Acute loss of Robo4 in R4cKO and R4ECcKO mice results in increased HSC mobilization after AMD3100 treatment compared to controls. No difference in WT BMC reconstitution is observed in Robo4cKO recipient mice that received 525 Rads radiation, while WT BMC reconstitution is reduced in Robo4cKO mice that received 260 Rads radiation. Overall, these results provide additional evidence that Robo4 is a key modulator of HSC location.

1. The authors show increased numbers of VECs in Robo4KO mice (Fig. 3B). Since the authors examined VCAM1 expression on SECs, the authors can show if the numbers of SECs are also increased in Robo4KO mice compared to WT.

2. It appears from the flow cytometry plots that the frequency of CD45 cells in WT(TAM+) group is significantly increased relative to the TAM-treated R4cKO and R4ECcKO mice. Based on total BM cellularity – is there a total reduction in CD45+ cells in the R4cKO and R4ECcKO mice compared to WT mice 3 days after TAM-treatment. If so, what are these cells and has this been previously reported?

3. Are the control flow cytometry plots shown in Fig 4A - Tam-treated WT mice after 3 days? The authors could provide the flow cytometry plots for the WT and R4KO mice that were not treated with TAM for comparison – even though similar data has been previously published. This would also be the same for Fig 4C.

4. Why did the authors choose to define immunophenotypic HSCs in the PB using CD27 (Lin-/CD27+/Kit+/Sca-1+/Flk2-)? Are there HSCs in the PB that lack CD27? Would they consider this an HSC-enriched population – if so, this should be defined in the text.

5. The authors need to provide some evidence that Robo4 is deleted in ECs in the Tam-treated R4ECcKO and R4cKO mouse models.

6. The authors conclude on lines 429-433 that homing and engraftment of Robo4-deficient BMC cannot be attributed to reduced levels of VCAM1..” While this conclusion seems reasonable for engraftment after acute loss, it does not preclude that VCAM1 could contribute to some of the Robo4-deficient engraftment after an extended period.

Reviewer #2: This is an interesting story by Smith-Berdan et al. that builds upon their previous publications demonstrating the importance of Robo4 in HSC mobilization and homing. The authors aimed to determine if global, germline deletion of Robo4 (previously published) is required for HSC trafficking compared to acute deletion in all cells or specifically in endothelium. Acute loss of Robo4 lead to alterations in HSC localization and HSC function without altering endothelial frequency and independent of VCAM1 expression. This is an exciting story that is well written and has scientific rigor. The data presented significantly adds more mechanism and insight into the role of Robo4 in HSC trafficking and location. Below are a few issues the reviewer would like to see addressed experimentally or discussed/clarified prior to publication.

Is it possible that R4KO mice are more protected following low dose radiation? Is there less space for the transplanted cells to engraft in R4KO mice following the myelosuppressive treatment? The images in Figure 2A indicate this could be the reason. Is it possible that the observed phenotypes in trafficking in R4KO mice is due to less damage for the surrounding tissue in the BM?

After preconditioning and long-term HSC transplants into R4KO recipients, have the authors checked to determine if the functional output of the HSCs the donor cells versus the residual endogenous HSCs (e.g., CD45.1 vs CD45.2) by performing secondary transplants. This is particularly interesting in the context of endothelial-specific deletion to exclude hematopoietic Robo4.

Are known paracrine factors that promote vascular health and HSC maintenance altered in acute and global R4KO mice (e.g., vegfr2, vegfr3, vegfa, bFgf, jagged 1 and 2, etc)?

Is there a difference in the mobilization efficiency following AMD3100 between 4 and 16 weeks post radiation?

6. PLOS authors have the option to publish the peer review history of their article (what does this mean?). If published, this will include your full peer review and any attached files.

Reviewer #1: No

Reviewer #2: **Yes: **Jason M Butler

---

## [Author Response · Author response to Decision Letter 0]

9 Jul 2021

Response to Reviewers' comments: 

We thank the Reviewers for the uniform “Yes” responses on the questions regarding the quality of our manuscript and for their suggestions for improvements to the body of work. We have addressed the feedback by addition of new experimental results or displays: Figure 2 (radiation sensitivity experiments), Figure 4A (new flow cytometry plots), Figure 4D (numbers of sinusoidal vascular endothelial cells (SECs) in the three different mouse models), Figure 5A (additional flow cytometry plots), Figure S1Di, Dii, and Diii (PCR and qRT-PCR from sorted VECs and HSCs to demonstrate Robo4 deletion), Figure S2C (numbers of CD45+ hematopoietic cells in the three different mouse models) and by making textual changes to clarify and better emphasize the points raised by the Reviewers. 

Reviewer #1: This manuscript extends the authors previous findings that Robo4 is extrinsically required for normal HSC trafficking. First, the authors confirm previous studies demonstrating that WT BMCs show decreased donor repopulation in Robo4KO recipients that receive a sublethal dose of radiation (535 Rads) compared to WT recipients and demonstrate no difference in the repopulation of WT BMCs in Robo4KO recipients that received high doses of radiation (725 and 1050 Rads). The authors demonstrate that Robo4KO mice show reduced sinusoidal area in BM sections stained with laminin compared to WT mice. Furthermore, while sinusoidal area is increased after irradiation, the sinusoidal area in Robo4KO mice is reduced compared to controls after 525 Rads radiation. However, while a difference in sinusoidal area is observed in mice that received 525 Rads, no difference in sinusoidal area is observed with increasing doses of irradiation (725 and 1050 Rads). The authors provide additional evidence for defective homing and extravasation of WT BMCs in Robo4KO mice in laminin-stained BM sections. The authors demonstrate that acute (3 day) loss of Robo4 does not affect total VEC cellularity, and VCAM1 expression on SECs, but does affect HSC mobilization in Rob4cKO and Robo4ECcKO mice. The authors provide data that acute loss of Robo4 in Robo4ECcKO mice results in increased vascular permeability. Acute loss of Robo4 in R4cKO and R4ECcKO mice results in increased HSC mobilization after AMD3100 treatment compared to controls. No difference in WT BMC reconstitution is observed in Robo4cKO recipient mice that received 525 Rads radiation, while WT BMC reconstitution is reduced in Robo4cKO mice that received 260 Rads radiation. Overall, these results provide additional evidence that Robo4 is a key modulator of HSC location.

1. The authors show increased numbers of VECs in Robo4KO mice (Fig. 3B). Since the authors examined VCAM1 expression on SECs, the authors can show if the numbers of SECs are also increased in Robo4KO mice compared to WT.

Response: Thank you for this suggestion. We have added the requested data as new Figure 4D (previous Figure 3 is now Figure 4). We found that germline, but not conditional, Robo4 deletion led to a significant reduction of SECs, defined as CD45-/Ter119-/CD31+/Sca1mid/Tie2+ BM cells. The germline Robo4 data are in agreement with our 2015 publication (Figures 4 A, D, and F). 

2. It appears from the flow cytometry plots that the frequency of CD45 cells in WT(TAM+) group is significantly increased relative to the TAM-treated R4cKO and R4ECcKO mice. Based on total BM cellularity – is there a total reduction in CD45+ cells in the R4cKO and R4ECcKO mice compared to WT mice 3 days after TAM-treatment. If so, what are these cells and has this been previously reported?

Response: The flow cytometry plots of the previous version made the BM cellularity appear different, but this was not a consistent or statistically significant observation. We have now replaced the flow cytometry plots with data that better represent the average outcome (Figure 4A) and the quantification showing lack of significant differences in CD45+ cells in the 3 mouse models (new Figure S2C).

3. Are the control flow cytometry plots shown in Fig 4A - Tam-treated WT mice after 3 days? The authors could provide the flow cytometry plots for the WT and R4KO mice that were not treated with TAM for comparison – even though similar data has been previously published. This would also be the same for Fig 4C.

Response: (Note: previous Figure 4 is now Figure 5) We have added the requested flow cytometry plots from the WT and R4KO mice for both the BM and PB HSC and MPP populations to Figure 5A. All TAM-treated animals were analyzed 3 days post their final injection of Tamoxifen (1 week post initial injection), as indicated in Figure S1B. 

4. Why did the authors choose to define immunophenotypic HSCs in the PB using CD27 (Lin-/CD27+/Kit+/Sca-1+/Flk2-)? Are there HSCs in the PB that lack CD27? Would they consider this an HSC-enriched population – if so, this should be defined in the text.

Response: We routinely use CD27 to eliminate the bright Lin-/cKit+/Sca1+ mast cell population that is abundant relative to HSCs in PB, but not BM. This practice is based in part on unpublished data and on this publication (now also referenced in the manuscript):

Vazquez SE, Inlay MA, Serwold T. CD201 and CD27 identify hematopoietic stem and progenitor cells across multiple murine strains independently of Kit and Sca-1. Exp Hematol. 2015;43: 578–585. doi:10.1016/j.exphem.2015.04.001

5. The authors need to provide some evidence that Robo4 is deleted in ECs in the Tam-treated R4ECcKO and R4cKO mouse models.

Response: We have added PCR data from sorted VEC and qRT-PCR data from both VEC and HSC showing genetic deletion of Robo4 and that the relative Robo4 mRNA levels were significantly reduced, cell type-selectively, in the Robo4KO, Robo4cKO, and Robo4ECcKO models compared WT controls (new Figure S1Di, ii and iii). 

6. The authors conclude on lines 429-433 that homing and engraftment of Robo4-deficient BMC cannot be attributed to reduced levels of VCAM1..” While this conclusion seems reasonable for engraftment after acute loss, it does not preclude that VCAM1 could contribute to some of the Robo4-deficient engraftment after an extended period.

Response: We agree with the Reviewer, and have added a statement (lines 415-416) to indicate this. 

Reviewer #2: This is an interesting story by Smith-Berdan et al. that builds upon their previous publications demonstrating the importance of Robo4 in HSC mobilization and homing. The authors aimed to determine if global, germline deletion of Robo4 (previously published) is required for HSC trafficking compared to acute deletion in all cells or specifically in endothelium. Acute loss of Robo4 lead to alterations in HSC localization and HSC function without altering endothelial frequency and independent of VCAM1 expression. This is an exciting story that is well written and has scientific rigor. The data presented significantly adds more mechanism and insight into the role of Robo4 in HSC trafficking and location. Below are a few issues the reviewer would like to see addressed experimentally or discussed/clarified prior to publication.

1. Is it possible that R4KO mice are more protected following low dose radiation? Is there less 

space for the transplanted cells to engraft in R4KO mice following the myelosuppressive treatment? The images in Figure 2A indicate this could be the reason. Is it possible that the observed phenotypes in trafficking in R4KO mice is due to less damage for the surrounding tissue in the BM?

Response: These are great questions that we have addressed with additional data from new experiments. We previously demonstrated that HSCs migrate poorly across Robo4-deficient endothelial cell layers in in vitro transwell assays (Smith-Berdan et al, 2015), indicating that trafficking across endothelial barriers is an important mechanism behind the poor engraftment in Robo4-deficient hosts. However, that does not preclude that differential radiation damage also play a role. We tested this by performing radiation sensitivity experiments, comparing cell numbers in sublethally irradiated WT to R4KO mice (new Figure 2; text on page 10). While the Reviewer hypothesized that mice lacking Robo4 would exhibit less damage to BM tissue, our data suggest the opposite outcome: there is significantly more damage to total cells, non-hematopoietic stromal cells, and VECs in the R4KO animals compared to the WT cohort (Figure 2B and D). This difference in radiation sensitivity was not a general property of Robo4-deficient cells, as HSCs, MPPs and myeloid progenitors displayed similar reductions upon irradiation in WT and R4KO mice (Figure 2A and C). Thus, while the quality of the space or niches was not addressed here, our data clearly show that improved survival of endothelial niche cells is not a reason for reduced engraftment in irradiated Robo4-deficient recipients. 

2. After preconditioning and long-term HSC transplants into R4KO recipients, have the authors checked to determine if the functional output of the HSCs the donor cells versus the residual endogenous HSCs (e.g., CD45.1 vs CD45.2) by performing secondary transplants. This is particularly interesting in the context of endothelial-specific deletion to exclude hematopoietic Robo4.

Response: If we understand the Reviewer’s comment correctly, the question is whether WT HSCs functionally decline in a R4-deficient environment. We are reluctant to perform these time- and resource-consuming experiments as we do not currently have evidence to suggest that from our current data: WT HSCs continue to produce cells at the same ratios as WT/WT for >16 wks in Fig 1B-D. We do appreciate the Reviewer’s suggestion and will keep the suggested approach in mind in the future. 

3. Are known paracrine factors that promote vascular health and HSC maintenance altered in acute and global R4KO mice (e.g., vegfr2, vegfr3, vegfa, bFgf, jagged 1 and 2, etc)?

Response: This is another good question that we are working on addressing. So far, we have focused on VCAM1 and also have additional negative data, but understanding the molecular interactions behind the phenotypes of Robo4 null mice is a high priority.

---

## [Editor Report · Decision Letter 1]

21 Jul 2021

Acute and endothelial-specific Robo4 deletion affect hematopoietic stem cell trafficking independent of VCAM1

PONE-D-21-02612R1

Dear Dr. Forsberg,

We’re pleased to inform you that your manuscript has been judged scientifically suitable for publication and will be formally accepted for publication once it meets all outstanding technical requirements.

Kind regards,

Kevin D Bunting

Academic Editor

PLOS ONE
---

## [Editor Report · Acceptance letter]

5 Aug 2021

PONE-D-21-02612R1 

Acute and endothelial-specific Robo4 deletion affect hematopoietic stem cell trafficking independent of VCAM1 

Dear Dr. Forsberg:

I'm pleased to inform you that your manuscript has been deemed suitable for publication in PLOS ONE. Congratulations! Your manuscript is now with our production department. 

Kind regards, 

on behalf of

Dr. Kevin D Bunting 

Academic Editor

PLOS ONE